# DiagrammerGPT: Generating Open-Domain, Open-Platform Diagrams via LLM Planning

**Abhay Zala**     **Han Lin**     **Jaemin Cho**     **Mohit Bansal**
UNC Chapel Hill
{aszala, hanlincs, jmincho, mbansal}@cs.unc.edu

https://diagrammerGPT.github.io

## Abstract

Text-to-image (T2I) generation has seen significant growth over the past few years. Despite this, there has been little work on generating diagrams with T2I models. A diagram is a symbolic/schematic representation that explains information using structurally rich and spatially complex visualizations (*e.g.*, a dense combination of related objects, text labels, directional arrows/lines, *etc*.). Existing state-of-the-art T2I models often fail at diagram generation because they lack fine-grained object layout control when many objects are densely connected via complex relations such as arrows/lines, and also often fail to render comprehensible text labels. To address this gap, we present DiagrammerGPT, a novel two-stage text-to-diagram generation framework leveraging the layout guidance capabilities of LLMs to generate more accurate diagrams. In the first stage, we use LLMs to generate and iteratively refine 'diagram plans' (in a planner-auditor feedback loop). In the second stage, we use a diagram generator, DiagramGLIGEN, and a text label rendering module to generate diagrams (with clear text labels) following the diagram plans. To benchmark the text-to-diagram generation task, we introduce AI2D-Caption, a densely annotated diagram dataset built on top of the AI2D dataset. We show that our DiagrammerGPT framework produces more accurate diagrams, outperforming existing T2I models. We also provide comprehensive analysis, including open-domain diagram generation, multi-platform vector graphic diagram generation, human-in-the-loop editing, and multimodal planner/auditor LLMs.

## 1 Introduction

Over the past few years, text-to-image (T2I) generation models (Rombach et al., 2021; Ramesh et al., 2022; Yu et al., 2022; Chang et al., 2023; Dai et al., 2023) have shown impressive advancements in image generation quality. Large language models (LLMs) have also recently shown strong capabilities and usefulness in broad language understanding and generation tasks (Touvron et al., 2023a;b; OpenAI, 2023b; Chung et al., 2022; Brown et al., 2020; Chowdhery et al., 2022). Recent works also have demonstrated that it is possible to leverage LLMs to control layouts for the downstream T2I models, for better semantic alignment with input text prompts (Cho et al., 2023b; Feng et al., 2023; Lian et al., 2023).

However, it has not been explored to use the combination of LLM and T2I generation frameworks for creating diagrams. A diagram is a symbolic/schematic representation that explains information using rich and spatially complex visualizations (*e.g.*, a dense combination of objects, text labels, arrows, lines, *etc*.). A system that helps to create accurate diagrams would be useful for preparing many educational and academic resources (*e.g.*, creating a diagram that explains new concepts in books, presentations, and papers). While the existing T2I generation models are good at generating realistic images, they often fail at diagram generation because they lack fine-grained object layout control when many objects are densely connected via complex relations such as arrows/lines and also often fail to render comprehensible text labels. In diagram generation, it is more important to

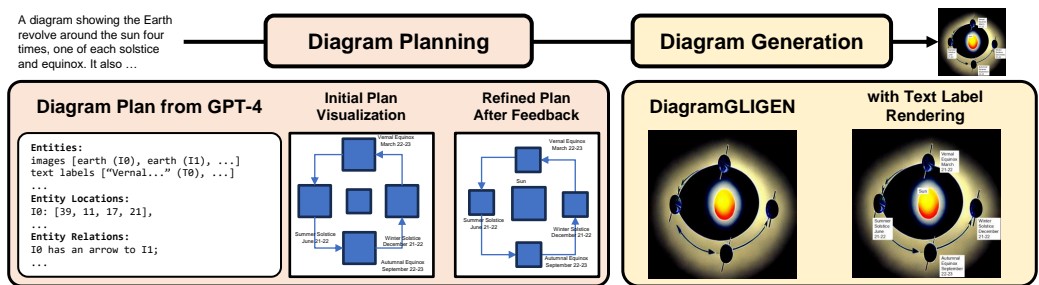

Figure 1: An overview of DiagrammerGPT, our two-stage framework for open-domain diagram generation. In the first diagram planning stage (Sec. 3.1), given a prompt, our LLM (GPT-4 (OpenAI, 2023b)) generates a *diagram plan*, which consists of dense entities, fine-grained relationships, and precise layouts. Then, the LLM iteratively refines the plan to correct mistakes. In the second diagram generation stage (Sec. 3.2), our DiagramGLIGEN outputs the diagram given the diagram plan, then, we render the text labels on the diagram.

convey correct information (with correct object relationships) than to generate photorealistic objects. In our experiments, existing T2I generation models usually generate diagrams where objects and arrows/lines have incorrect relationships and rendered text labels are incomprehensible. Even the recent state-of-the-art (SOTA) model DALL-E 3 (OpenAI, 2023a) struggles to render accurate diagrams (mention in their system card and shown in Fig. 6).

To address the issues in text-to-diagram generation, we introduce DiagrammerGPT, a novel two-stage framework capable of generating more accurate open-domain diagrams for multiple platforms by following LLM-generated diagram plans. As shown in Fig. 1, our DiagrammerGPT splits the text-to-diagram generation task into two stages: **diagram planning** and **diagram generation**. For the first stage (Fig. 1 left), we employ an LLM to create and refine '*diagram plans*'. In the second stage (Fig. 1 right), we introduce DiagramGLIGEN, a layout-guided diagram generation module, to generate diagrams based on diagram plans, then explicitly render text labels, ensuring their readability.

In the first stage, diagram planning (Sec. 3.1), we employ an LLM (*e.g.*, GPT-4 (OpenAI, 2023b)) to act as a *planner* and generate **diagram plans** given text prompts. The diagram plans consist of (1) a list of entities (*i.e.*, objects and text labels), (2) relationships between the entities (*i.e.*, arrows or lines between entities), and (3) entity layouts (*i.e.*, 2D bounding box coordinates). After the initial generation of diagram plans, inspired by recent works on LLMs that self-refine previously generated contents (Chen et al., 2023c; Miao et al., 2023; Madaan et al., 2023), we use another LLM to act as an *auditor* and find potential errors such as incorrect object positions or relationships between objects. Then, the planner LLM takes feedback from the auditor LLM to update the diagram plan (*e.g.*, in Fig. 1, the sun is smaller than the earths, so the plan is adjusted to fix the relative scales). We find that our LLM-generated diagram plans are quite accurate and that refinement can effectively help correct some small errors (see Sec. 5.2 and feedback example in appendix for more details).

In the second stage, diagram generation (Sec. 3.2), we introduce DiagramGLIGEN, a layout-guided diagram generation module, to generate diagrams based on diagram plans, then explicitly render text labels, ensuring their readability. We implement DiagramGLIGEN based on GLIGEN (Li et al., 2023b) architecture, which adds gated self-attention layers to the Stable Diffusion v1.4 (Rombach et al., 2021) model. While the original GLIGEN model is only trained on natural images and take only objects for layout grounding, DiagramGLIGEN is more specialized in the diagram domain, by being trained on our new AI2D-Caption diagram dataset (see the following paragraph and Sec. 4.1 for more details) and taking the text labels and arrows as additional layout grounding inputs. As the SOTA diffusion models still struggle in rendering text (Liu et al., 2023b; Chen et al., 2023a; Cho et al., 2023b), we explicitly render text labels on top of the generated diagrams to ensure they are readable. Compared to the T2I baselines, our diagram generation stage allows more accurate object layouts and relationships between the objects (*e.g.*, arrows/lines), and clear text labels.

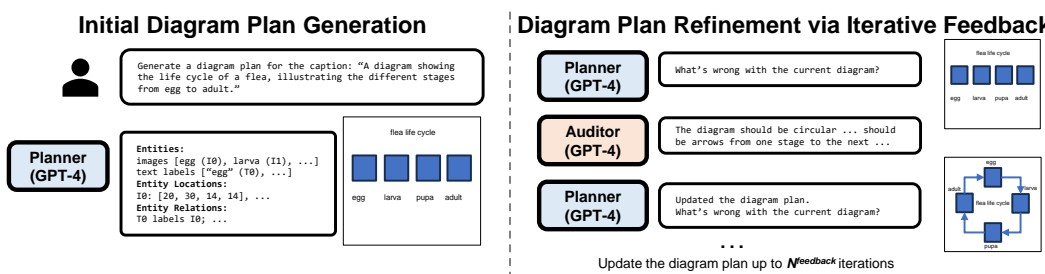

Figure 2: Illustration of the first stage of DiagrammerGPT: diagram planning (Sec. 3.1). We use a planner LLM (*e.g.*, GPT-4 (OpenAI, 2023b)) to create the fine-grained layouts of diagrams, which we call *diagram plans*. We first generate an initial diagram from the input text prompt with an LLM (**left**). Then we iteratively refine diagram plans in a feedback loop of the planner and auditor LLMs.

Since there are no diagram datasets with detailed captions and fine-grained layout annotations, we construct **AI2D-Caption**, a new dataset for the text-to-diagram generation task built on top of the AI2D (Kembhavi et al., 2016) dataset (Sec. 4.1). We employ LLaVA 1.5 (Liu et al., 2023a), a SOTA multimodal LLM, to create annotations of captions and object descriptions on AI2D diagrams.

We comprehensively compare our DiagrammerGPT to recent text-to-image/diagram generation methods, including Stable Diffusion v1.4 (Rombach et al., 2021), VPGen (Cho et al., 2023b), and AutomaTikZ (Belouadi et al., 2023), in both zeroshot and fine-tuned settings (see Sec. 4.2 and Sec. 5.1). In our quantitative and qualitative analysis, our DiagrammerGPT demonstrates more accurate diagram generation performance than the baseline models. Our method also outperforms Stable Diffusion v1.4, the closest and strongest baseline, in a human preference study on both image-text alignment and object relationship criteria. In our error analysis, we show that diagram plans generated by our LLM (GPT-4) are quite accurate (before and after refinement), while DiagramGLIGEN sometimes makes mistakes when generating diagrams. This indicates that our DiagrammerGPT can benefit from future layout-guided image generation backbones stronger than GLIGEN (see Sec. 5.2).

We also conduct additional analysis, including **open-domain** diagram generation, **multiple platforms vector graphic** diagram generation, **human-in-the-loop** editing, and **text-only vs. multimodal LLM** (*e.g.*, GPT-4Vision (OpenAI, 2023c); see Sec. 5.4 and appendix). First, we experiment with generating diagrams in unseen domains (*e.g.*, geology and plants) that are not covered in the LLM in-context learning domains (astronomy, biology, engineering), where our DiagrammerGPT could often generate semantically accurate diagram plans and diagrams from these unseen prompts. Second, we experiment with rendering our diagram plans in multiple platforms: Microsoft PowerPoint, Inkscape, and Adobe Illustrator (see Fig. 7). Third, we show that once a diagram plan is exported to another platform, end-users can also manually edit plans to their liking and send them back to our DiagramGLIGEN to generate images using their manually refined plan (*i.e.*, human-in-the-loop refinement of diagram plan). Lastly, we experiment with using the recently released GPT-4Vision (OpenAI, 2023c) instead of text-only GPT-4 for our planner and auditor LLMs (in appendix).

## 2 Related Works

### 2.1 Text-to-Image Generation

In the text-to-image (T2I) generation task, models generate an image from a given text prompt. Recently multimodal language models (*e.g.*, Parti (Yu et al., 2022) and MUSE (Chang et al., 2023)) and diffusion models (*e.g.*, Stable Diffusion (Rombach et al., 2021), DALL-E 2 (Ramesh et al., 2022), and Imagen (Saharia et al., 2022)) have gained popularity for this task. These recent T2I generation models have demonstrated impressive photorealism in their zeroshot image generation capabilities. However, using these models directly for diagram

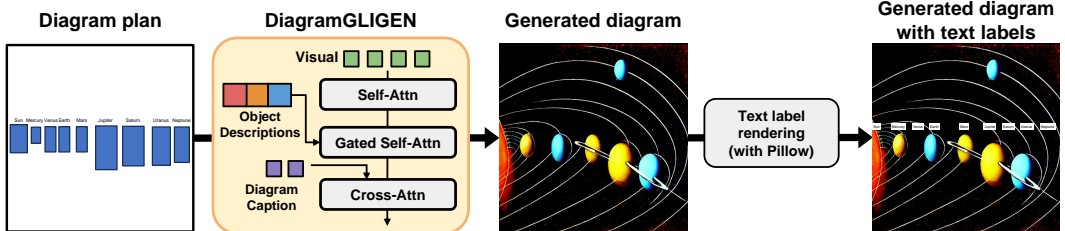

Figure 3: Illustration of the second stage of DiagrammerGPT: diagram generation (Sec. 3.2). We first generate the objects from the diagram plan with DiagramGLIGEN, our layout-guided diagram generation model. Then, we use Pillow to render clear text labels.

generation is challenging due to the significant distribution gap between diagrams and their training images. Additionally, these models often lack precise control of fine-grained layouts when many objects are densely connected via complex relations and frequently produce illegible text labels, which are essential for the diagram generation task.

## 2.2 Text-to-Image Generation with LLM-Guided Layouts

Large language models (LLMs) have demonstrated their usefulness in various language generation tasks (Touvron et al., 2023a;b; OpenAI, 2023b; Chung et al., 2022; Brown et al., 2020; Chowdhery et al., 2022). Recent works also leverage LLMs to control layouts for the downstream image generation models, for better semantic alignment with input text prompts (Cho et al., 2023b; Feng et al., 2023; Lian et al., 2023). However, these works focus on generating natural images and do not have capabilities that are crucial in diagram generations, *e.g.*, rendering clear text labels or the ability to precisely control fine-grained layouts of many objects that are densely connected via complex relations such as arrows/lines, as shown in our experiments (Sec. 5). A concurrent work, AutomaTikZ (Belouadi et al., 2023) uses LLMs to generate TikZ (Tantau, 2007)[1] code to produce scientific vector graphics. While TikZ can be used to draw specific types of diagrams, such as 2D bar plots or directional acyclic graphs, it is difficult to generate diagrams including entities not supported by the TikZ primitives, such as animals. In contrast, our DiagrammerGPT allows for generating diagrams including diverse entities (*e.g.*, different animals in life cycle diagrams or different planets/stars in astronomy diagrams).

## 3 DiagrammerGPT: Method Details

We introduce DiagrammerGPT, a novel two-stage framework for generating open-domain diagrams from text prompts, where an LLM first generates the overall plan, and a visual generator renders an actual diagram following the plan. In the first stage, **Diagram Planning** (Sec. 3.1), a *planner* LLM takes a text description of a diagram as input and generates a *diagram plan*, an overall diagram layout that guides the downstream diagram generation module. The *planner* LLM generates the initial diagram plan and iteratively refines the diagram plan via feedback from an auditor LLM. In the second **Diagram Generation** stage (Sec. 3.2), DiagramGLIGEN, our new layout-guided diagram generation module, takes the diagram plan and generates the diagram. Finally, we render the text labels (which are from the diagram plan) onto the diagram to ensure clear and easy-to-read labels for each entity.

## 3.1 Stage 1: Diagram Planning

As illustrated in the left part of Fig. 2, we use an LLM (e.g., GPT-4 (OpenAI, 2023b)) to create the overall layouts for diagrams, which we call *diagram plans* (see appendix for detailed plan configuration and example).

---

[1]A TeX package for generating graphics by composing primitive objects with basic polygons.

**Initial diagram plan generation.** We generate diagram plans with GPT-4 via 10 in-context learning examples. Plans consist of three components: (1) entities - a dense list of objects; (2) relationships - complex relationships between entities; and (3) layouts - 2D bounding boxes of the entities. See appendix for diagram plan generation prompts/extra details.

**Diagram plan refinement via iterative feedback.** Although the *planner* (GPT-4) generates fairly accurate initial diagram plans, we can further refine it to account for potentially missing or improperly arranged entities. To address this issue, we introduce an *auditor* LLM that checks for any mismatch between the current diagram plan and the input prompt. It then provides feedback, enabling the *planner* LLM to refine the diagram plans. Our *auditor* and *planner* LLMs form a feedback loop to iteratively refine the diagram plans. For this *auditor* LLM, we employ GPT-4 again but with a different preamble and in-context examples designed to give useful feedback (see appendix for the prompts to initialize the *auditor* LLM). We repeat the feedback loop for up to $N$ iterations, using $N = 4$ in our experiments. The right part of Fig. 2 exemplifies the feedback process.

### 3.2 Stage 2: Diagram Generation

As shown in Fig. 3, we first generate the diagram images following the diagram plan with DiagramGLIGEN, then render text labels on the diagram.

**DiagramGLIGEN: Layout-guided diagram generation.** Conveying factual information is more crucial than drawing photorealistic objects in diagram generation. In our experiments, we observe that existing T2I models often omit important objects, and generate incorrect object relationships and unreadable text labels (see Sec. 5 and Fig. 5). Therefore, we introduce DiagramGLIGEN, a layout-guided text-to-diagram generation model to tackle these issues. Inspired by GLIGEN (Li et al., 2023b), we implement DiagramGLIGEN by incorporating gated self-attention layers, which take layout grounding inputs, into Stable Diffusion v1.4. Furthermore, we enhance layout control by incorporating text labels and relationships as part of the layout grounding inputs during training, which can reduce the generation of unreadable text and redundant arrows/lines during inference (by not giving text labels during inference). We employ CLIP (Radford et al., 2021) text encoder to encode object descriptions and their relationships. We use the CLIP image encoder to represent bounding box regions of the text labels in the ground truth diagrams. See the appendix for additional GLIGEN details and training setup.

**Text label rendering.** Text labels (*e.g.*, "Sun" labeling the sun object in Fig. 3) in diagrams can effectively assist readers in understanding new concepts (Johnson et al., 2014). However, as shown in Cho *et al.* (Cho et al., 2023b), existing T2I generation models (including Stable Diffusion v1.4) still struggle to generate high-quality text labels. Therefore, we explicitly render clear text labels on the diagrams with the Pillow Python package (Clark, 2015).

## 4 Experimental Setup

In the following subsections, we introduce our AI2D-Caption dataset for the text-to-diagram generation task (Sec. 4.1), baseline models (Sec. 4.2), evaluation metrics (Sec. 4.3), and human evaluation setups (Sec. 4.4).

### 4.1 AI2D-Caption Dataset

We introduce the AI2D-Caption dataset for the text-to-diagram generation task. AI2D-Caption is built on top of AI2 Diagrams (AI2D) dataset (Kembhavi et al., 2016), which provides annotations of around 4.9K diagrams covering diverse scientific domains, from Grade 1-6 science textbooks. The original AI2D dataset's annotations are very short / are missing some aspects (*i.e.*, each object bounding box is labeled simply as 'blob'), so we employ LLaVA 1.5 (Liu et al., 2023a), a SOTA multimodal language model, to generate detailed captions for the diagrams and region descriptions of each bounding box (see Fig. 4 for comparison of AI2D and AI2D-Caption and the appendix for implementation details).

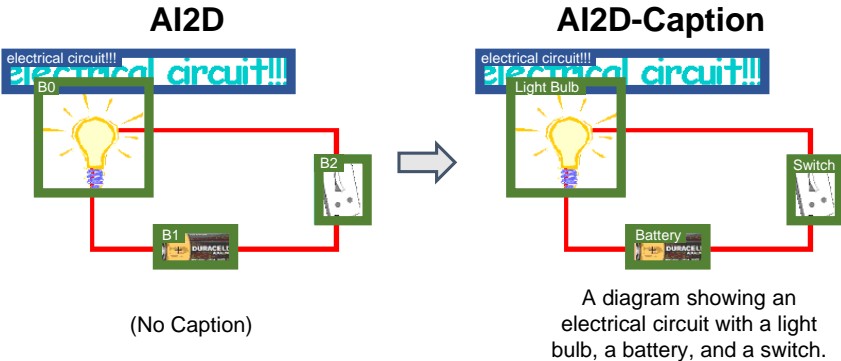

Figure 4: Example diagram annotation from the AI2D dataset (Kembhavi et al., 2016) (left) and our AI2D-Caption (right). AI2D-Caption additionally provides annotations of the diagram caption and bounding box region descriptions.

We use all of the LLaVA 1.5-aided caption and region annotations for DiagramGLIGEN training and baseline fine-tuning.

To ensure that the LLM in-context examples and evaluation are accurate, we manually annotate diagram plans (*i.e.*, captions, object/text label bounding boxes, object descriptions, arrows) for randomly selected 105 diagrams (held out from the training set described above). Among the 105 diagrams, we use 30 diagrams that cover diverse scientific domains (10 for astronomy, 10 for biology, and 10 for engineering) as in-context examples, and 75 diagrams (25 for astronomy, 25 for biology, and 25 for engineering) as a test split.

## 4.2   Baseline Models

We compare our DiagrammerGPT to several baseline models, including Stable Diffusion v1.4 (Rombach et al., 2021), VPGen (Vicuna13B (Chiang et al., 2023) + GLIGEN) (Cho et al., 2023b), and AutomaTikZ (Belouadi et al., 2023) (CLiMA-13B). For the baselines, we experiment with zero-shot and fine-tuned (on our AI2D-Caption dataset where applicable) diagram generation. For VPGen, we fine-tune both the Vicuna13B and GLIGEN.

## 4.3   Evaluation Metrics

**VPEval (objects, counts, relationships, texts).** The VPEval metric (Cho et al., 2023b) evaluates diagrams in terms of the presence of objects (object evaluation), the number of objects (count evaluation), the correctness of spatial and connection relationships (relationship evaluation), and the presence of correct text labels (text evaluation). While the original VPEval uses BLIP-2 (Li et al., 2023a) as its visual reasoning model, we employ the recently released LLaVA 1.5 (Liu et al., 2023a) model as our visual reasoning model, as we find it is more faithful in diagram question answering in our initial experiments and is an overall stronger model than BLIP-2 (Liu et al., 2023a). See appendix for setup details.

**Captioning.** In line with previous works (Hong et al., 2018; Hinz et al., 2020; Cho et al., 2023a), we also use captioning as a way to determine the accuracy of the generated diagrams. We use LLaVA 1.5 (Liu et al., 2023a) to caption each generated diagram and then compare this generated caption with the ground-truth caption using CIDEr (Vedantam et al., 2014) and BERTScore (Zhang et al., 2020).

**CLIPScore.** Following previous works (Cho et al., 2023a; Saharia et al., 2022; Belouadi et al., 2023), we use CLIPScore (Hessel et al., 2021) to measure the similarity between the generated diagram and the original caption/ground-truth diagram. Concretely, we calculate two types of CLIPScore: (1) CLIPScore[Img-Txt]: similarity between the generated diagram and

| Methods | VPEval (%) ↑ | | | | | Captioning ↑ | | CLIPScore ↑ | |
|---|---|---|---|---|---|---|---|---|---|
| | Object | Count | Text | Relationships | Overall | CIDEr | BERTScore | Img-Txt | Img-Img |
| *Zeroshot* | | | | | | | | | |
| Stable Diffusion v1.4 | 63.1 | 27.8 | 0.0 | 79.3 | 39.0 | 7.7 | 87.5 | 27.3 | 65.3 |
| VPGen | 55.8 | 32.9 | 0.0 | 72.8 | 39.3 | 6.1 | 87.2 | 25.6 | 61.7 |
| AutomaTikZ | 24.9 | 34.2 | 5.5 | 67.7 | 31.0 | 12.2 | 86.9 | 24.7 | 64.5 |
| *Fine-tuned* | | | | | | | | | |
| Stable Diffusion v1.4 | 69.8 | 35.4 | 0.0 | 81.9 | 45.1 | 18.2 | 88.5 | 30.1 | 68.1 |
| VPGen | 62.8 | 27.8 | 0.0 | 76.3 | 39.7 | 4.2 | 86.9 | 26.4 | 61.9 |
| DiagrammerGPT (Ours) | **86.4** | **57.0** | **47.5** | **87.9** | **71.2** | **26.4** | **89.4** | **32.1** | **73.9** |

Table 1: Comparison of DiagrammerGPT to existing text-to-image generation baseline models. On all metrics, DiagrammerGPT outperforms the baselines, indicating that our method is more effective for generating accurate diagrams.

the ground-truth caption. (2) CLIPScore[Img-Img]: similarity between the generated diagram and the ground-truth diagram. We use OpenAI CLIP ViT-L/14 (Radford et al., 2021).

## 4.4 Human Evaluation

**Human error analysis of the two stages.** As our DiagrammerGPT pipeline consists of two stages, it is important to understand where any errors in our pipeline come from. In both stages, we assess (1) *Object Presence*: whether all required objects for the diagrams are present, and (2) *Object Relationships*: if the objects have proper relationships to each other (*e.g.*, for a lunar eclipse, the earth should be between the sun and the moon). We have an expert rate both stages on a Likert scale from 1 to 5 for 25 layouts/images.

**Pairwise preference: Image-text alignment & object relationships.** We conduct a human analysis comparing our DiagrammerGPT framework to Stable Diffusion v1.4 on 50 diagrams. We choose Stable Diffusion v1.4 fine-tuned on AI2D-Caption for comparison because it is the closest baseline to our DiagrammerGPT and also shows the strongest results among the baselines (see Sec. 5.1). We ask crowd-sourced annotators (20 unique annotators) from Amazon Mechanical Turk[2] to evaluate the diagrams generated for each prompt (see appendix for setup details).

## 5 Results and Discussion

We show our primary quantitative results (Sec. 5.1), human evaluation on pairwise preference study and error analysis (Sec. 5.2), qualitative analysis (Sec. 5.3), additional analysis about open-domain generation, multi-platform vector graphic diagram generation, human-in-the-loop diagram plan editing, (Sec. 5.4), and an ablation of different ablations for diagram plan generation (Sec. 5.5). See appendix for more analysis on DiagramGLIGEN setups and multimodal planner/auditor LLMs.

## 5.1 Quantitative Results

**VPEval.** Table 1 left block shows the VPEval results. For both Stable Diffusion v1.4 and VPGen baselines, fine-tuning improves the score for object skill (*e.g.*, 63.1 → 69.8 for Stable Diffusion v1.4, and 55.8 → 62.8 for VPGen), but for VPGen count, it decreases scores (32.9 → 27.8). For relationships, both models improve (79.3 → 81.9 for Stable Diffusion v1.4, and 72.8 → 76.3 for VPGen). For text, both models achieve 0 scores before and after fine-tuning. Our DiagrammerGPT outperforms both zeroshot and fine-tuned baselines on both overall and skill-specific VPEval scores, showcasing the strong layout control, object relationship representation, and accurate text rendering capability of our diagram generation framework.

**Captioning and CLIPScore.** Table 1 middle block shows captioning scores (with LLaVA 1.5), and the right block shows CLIPScore (with CLIP-ViT L/14). Our DiagrammerGPT

---

[2]https://www.mturk.com

| Stage 1: Diagram Planning (with GPT-4) | | | | Stage 2: Diagram Generation (with DiagramGLIGEN) | |
|---|---|---|---|---|---|
| Initial diagram plan | | Diagram plan after refinement | | Final Diagram | |
| Objects Presence (↑) | Object Relations (↑) | Objects Presence (↑) | Object Relations (↑) | Objects Presence (↑) | Object Relations (↑) |
| 4.96 | 4.56 | 4.96 | 4.72 | 2.96 | 3.36 |

Table 2: Step-wise error analysis of DiagrammerGPT on 25 AI2D-Caption test prompts. We use a Likert scale (1-5) to evaluate object presence and object relations of the diagram plan (before and after refinement) and the final generated diagram.

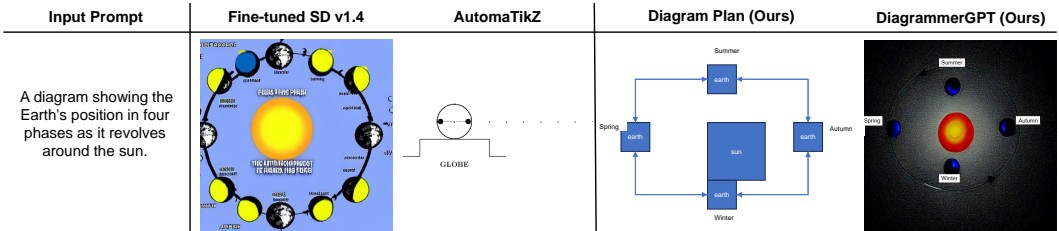

Figure 5: Example diagram generation results from baselines (fine-tuned Stable Diffusion v1.4 and AutomaTikZ) and our DiagrammerGPT on the AI2D-Caption test split. Our DiagrammerGPT correctly follows the caption while the baselines make several errors.

outperforms both the zeroshot and fine-tuned baselines, indicating our generated diagrams have more relevant information to the input prompt (which is a critical aspect of diagrams), have better image-text alignment, and more closely resemble the ground-truth diagrams (image-image) than the baselines. Our DiagrammerGPT significantly outperforms both fine-tuned VPGen (26.4 *vs.* 4.2) and fine-tuned Stable Diffusion v1.4 (26.4 *vs.* 18.2) for CIDEr and also achieves a few higher points on BERTScore. For both CLIPScore's, DiagrammerGPT has improvement over fine-tuned Stable Diffusion v1.4 (32.1 *vs.* 30.1 and 73.9 *vs.* 68.1).

## 5.2 Human Evaluation

**Step-by-step error analysis.** Table 2 shows that our diagram plans exhibit high scores on both object presence (4.96) and object relationship (4.56) even before the refinement, and refinement increases the object relationship scores even further (4.56 → 4.72) by adjusting the entity layouts from the initial diagram plan (see appendix for refinement examples). During the diagram generation stage, the object presence and relationship scores decrease. This indicates that our DiagrammerGPT could generate more accurate diagrams, once we have access to a layout-guided image generation backbone model stronger than Stable Diffusion v1.4 architecture.

**Human preference.** We conduct a human preference study, comparing our DiagrammerGPT and fine-tuned Stable Diffusion v1.4 (SD v1.4) in image-text alignment and object relationships. As shown in Table 3, our DiagrammerGPT achieves a higher preference than Stable Diffusion v1.4 in both image-text alignment (68% vs 24%) and object relationships (62% vs 34%) criteria.

| Evaluation category | Human Preference (%) ↑ | | |
|---|---|---|---|
| | DiagrammerGPT | SD v1.4 | Tie |
| Image-Text Alignment | **68** | 24 | 8 |
| Object Relationships | **62** | 34 | 4 |

Table 3: Human preference study on generated diagrams: DiagrammerGPT *vs.* SD v1.4.

## 5.3 Qualitative Analysis

**Comparison with baselines.** Fig. 5 shows example diagrams generated by the baselines (Stable Diffusion v1.4 and AutomaTikZ) and our DiagrammerGPT (both diagram plan and final generation diagram) on the AI2D-Caption test split (see appendix for additional example). Our diagram plan strongly reflects the prompt, and the final diagram is more aligned with the input prompt than the baselines. In the Fig. 5 example, our diagram

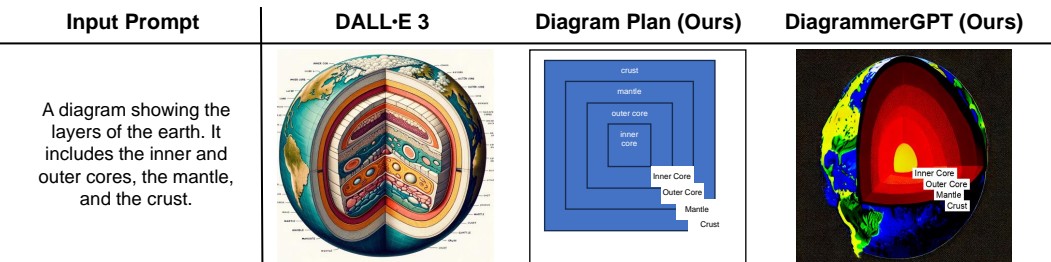

Figure 6: Open-domain diagram generation examples with DALL-E 3 and DiagrammerGPT.

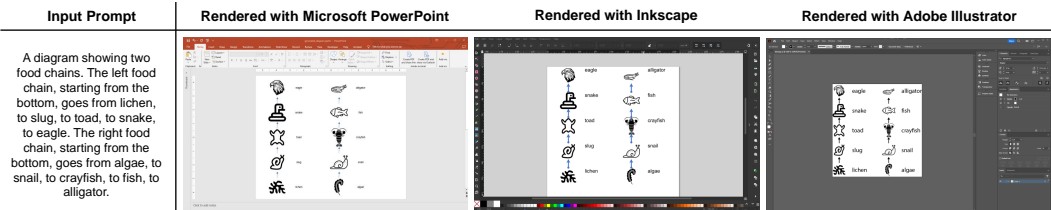

Figure 7: Examples of vector graphic diagrams generated from diagram plans.

correctly shows the earth in four phases revolving around the sun. Stable Diffusion v1.4 either over-generates objects in the image (*e.g.*, too many earths), and AutomaTikZ fails to generate a proper diagram. Although our generated diagram plans are generally correct, sometimes DiagramGLIGEN can fail to properly follow all aspects. As noted in Sec. 5.2, our DiagramGLIGEN can improve once a better backbone becomes available.

**Diagram plan refinement.** We analyze how our diagram refinement step (see Sec. 3.1) improves the diagram plans. The refinement can fix minor mistakes like missing connections in a circuit or objects being placed in the wrong location. We show examples in the appendix.

### 5.4 Additional Analysis: Open-Domain, Open-Platform, and Human-in-the-Loop

**Open-domain diagram generation.** We demonstrate that our planner LLM can extend its capabilities to unseen domains beyond the three areas (astronomy/biology/engineering) given in the in-context examples (30 total examples, 10 from each of astronomy/biology/engineering). As shown in Fig. 6, from an input prompt in the earth science domain, our planner LLM generates fairly accurate layouts, and our DiagramGLIGEN can generate a diagram following the layouts. We also compare the recently released DALL-E 3 (OpenAI, 2023a) model and find that it generally produces images with good aesthetic style but tends to generate diagrams with redundant objects (*e.g.*, excessive text descriptions or objects). It also struggles with creating diagrams that adhere to a prompt (*e.g.*, generating incorrect layers in the earth example). The DALL-E 3 system card (OpenAI, 2023a) also notes that DALL-E 3 tends to generate inaccurate information in diagrams. See appendix for more examples.

**Vector graphic diagram generation in different platforms.** Although our primary focus is on a pixel-level diagram generation pipeline with DiagramGLIGEN, our diagram plans can also be used for vector graphic diagrams, with multiple platforms like Microsoft PowerPoint, Inkscape, and Adobe Illustrator. Fig. 7 presents an example of vector graphic diagrams. The example delivers promising results by effectively conveying the crucial information and layouts described in the input text prompts (see additional example in the appendix). However, it also exhibits certain limitations: (1) inconsistency in icon styles; (2) limited icon retrieval capability. As the diagrams are editable via these same platforms, these limitations can be addressed by end-users editing the diagrams to their liking (see below).

**Human-in-the-loop diagram plan editing.** With the diagram plans being rendered in vector graphic platforms, as mentioned above, our DiagrammerGPT can provide an editable dia-

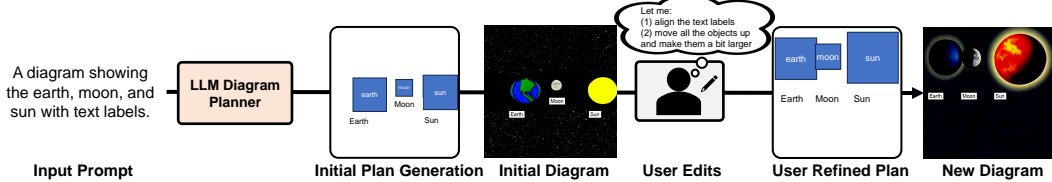

Figure 8: Illustration of human-in-the-loop diagram plan editing. DiagrammerGPT first provides an initial diagram plan with the corresponding generated diagram, users can then review the generated layouts/diagrams and make adjustments based on their needs.

gram plan, allowing for human-in-the-loop editing. As illustrated in Fig. 8, DiagrammerGPT first generates an initial diagram plan along with the rendered image. Users can then review the generated layouts/diagrams and make adjustments based on their needs/wants (*e.g.*, move the objects, add/remove objects, adjust object sizes, etc.). With the human-refined diagram plan, users can either keep it in vector format and use icons (as mentioned in the previous paragraph) or give it back to DiagramGLIGEN and then create pixel-level diagrams, resulting in diagrams/layouts that are better suited to end-users requirements.

## 5.5 Different LLMs for diagram plan generation

We also experiment with different LLMs such as GPT-3.5 Turbo (OpenAI, 2023d), GPT-4 (OpenAI, 2023b), LLaMA 3 (8B) (AI@Meta, 2024), and LLaMA2-Chat (13B) (Touvron et al., 2023b). We compare them with 'layout recall' – how often each ground-truth object/text label can be found in the predicted diagram plan. We first obtain bipartite matching of objects/text labels between prediction and ground-

| LLM | Layout Recall | | |
|---|---|---|---|
| | Object | Text | Overall |
| LLaMA3 8B | 24.9 | 32.2 | 29.2 |
| VPGen (Vicuna 13B) | 30.6 | 0 | 12.9 |
| GPT-3.5 Turbo | 82.5 | 54.6 | 74.7 |
| GPT-4 (default) | **84.1** | **60.1** | **78.4** |

Table 4: Ablation of different LLMs for the planner and auditor.

truth via Hungarian matching algorithm, using BLEU-1 (Papineni et al., 2002) as the matching score. Then we calculate layout recall metric: $\frac{\sum_{n=1}^{N} match(obj_n)}{N}$, where $N$ is the total number of ground-truth object/text labels, $obj_n$ is nth the ground-truth object/text label, and *match* outputs 1 if the object/text label is matched in the predicted diagram plan and 0 if not.

Table 4 shows a comparison of four LLMs: LLaMA3 8B, GPT-3.5 Turbo, GPT-4, and Vicuna 13B (VPGen checkpoint). As indicated by GPT-3.5 Turbo's and GPT-4's high 'overall' performance (74.7 and 78.4 respectively), both models are capable of generating accurate diagrams; however, as GPT-4 is slightly better in both the 'object' and 'text' metrics, we use GPT-4 as our main LLM. LLaMA3 and fine-tuned Vicuna are not able to perform well for the task with both getting an overall score below 30.

## 6 Conclusion

In this work, we propose DiagrammerGPT, a novel two-stage text-to-diagram generation framework that leverages the knowledge of LLMs for planning and refining the overall diagram plans. We demonstrate that our DiagrammerGPT achieves more semantically accurate layouts in diagram generation than baseline models in both quantitative and qualitative analysis. In addition, we provide comprehensive human error analysis, ablation studies, and analysis about open-domain diagram generation, vector graphic diagram generation, human-in-the-loop diagram plan editing, and multimodal planner/auditor LLMs. We hope our work can inspire future research on diagram generation.

## Acknowledgments

We thank the reviewers for the thoughtful discussion and feedback. This work was supported by DARPA ECOLE Program No. HR00112390060, NSF-AI Engage Institute DRL-2112635, DARPA Machine Commonsense (MCS) Grant N66001-19-2-4031, ARO Award W911NF2110220, ONR Grant N00014-23-1-2356, Accelerate Foundation Models Research program, and a Bloomberg Data Science Ph.D. Fellowship. The views contained in this article are those of the authors and not of the funding agency.

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

# Appendix

In this appendix, we provide LLM prompt templates and diagram plans used in the diagram planning stage (Appendix A), AI2D-Caption collection details,(Appendix B), additional DiagramGLIGEN details (Appendix C), experimental setup details (Appendix D), human evaluation setup details (Appendix E), additional results/analysis/visualizations/ablations details (Appendix F), and limitations (Appendix G).

## A    LLM Prompt Templates and Diagram Plans

**Prompt templates.**    In Fig. 9 and Fig. 10, we show the prompt templates for the planner LLM and auditor LLM, respectively. For both LLMs we provide 10 in-context examples followed by the inputs. For planner LLM, we give it the caption and the topic of the caption (*e.g.*, astronomy, biology, *etc.*). For auditor LLM, we give the diagram plan generated by the planner and ask if there are any issues.

**Diagram plans.**    A *diagram plan* consists of three components: (1) entities - a dense list of objects (*e.g.*, larva in Fig. 2) and text labels (*e.g.*, "egg" in Fig. 2); (2) relationships - complex relationships between entities (*e.g.*, object-object relationship "[obj_0] *has an arrow to* [obj_1]" or object-text label relationship "[text_label_0] *labels* [obj_0]"); (3) layouts - 2D bounding boxes of the entities (*e.g.*, "[obj_0]: $[20, 30, 14, 14]$" in Fig. 2). For object-object relationships, we utilize two types: line and arrow (a line with explicit start and end entities), which are useful when specifying object relationships in diagrams such as flow charts or life cycles. For object-text label relationships, we specify which object each label refers to. For layouts, we use the $[x, y, w, h]$ format for 2D bounding boxes, whose coordinates are normalized and integer-quantized within $\{0, 1, \cdots 100\}$, in accordance with VPGen (Cho et al., 2023b).

**Full diagram plan example.**    Fig. 11 shows an example of a fully generated diagram plan by our LLM planner for the prompt "A diagram showing the life cycle of a butterfly, going from an egg to larva to pupa to an adult butterfly and repeating." This diagram plan corresponds to the second example in Fig. 13.

**LLM API Costs**    The average input token length for the planner stands at 4.6K, while the average output token length is 0.5K. Generating a diagram plan using GPT-4 costs $0.17 USD.

## B    AI2D-Caption Collection Details

To create the AI2D-Caption dataset described in main paper Sec. 4.1, we employ LLaVA 1.5 (Liu et al., 2023a), a state-of-the-art multimodal language model, to generate captions and bounding box region descriptions in AI2D diagrams. The original AI2D dataset provides annotations for diagrams, including titles, bounding boxes for object/text labels, and object linkages (*e.g.* arrows/lines between objects). However, since the dataset is designed for

**Instructions:**
Given a caption of a diagram and topic, generate the diagram layout, and then a list of required entities that would be needed to create the described diagram. Then generate a list of the relationships between the entities (i.e. which ones are connected or labeling each other). Finally, generate the location of each entity.
An entity can be an image or text. Entity locations should be generated in [x, y, width, height] format, where 0,0 is the top left corner and 100,100 is max image size.

Think step-by-step, break each caption into parts and generate the required entities, relationships, and locations for each part.

Here are some rules to follow:
All numbers should be positive, do not generate negative numbers.
Please always generate a list of entities, even if the list is empty.
Entities should not be outside the bounds [0, 0, 100, 100].
The x coordinate + the width of an entity should not exceed 100.
The y coordinate + the height of an entity should not exceed 100.
Entities of the same type should not overlap.

**In-context examples:**
Caption:
A diagram showing the positions of the Earth, the Moon, and the Sun during a lunar eclipse. The Earth is directly between the sun and the moon. The moon is in the earth's shadow.
Topic:
…
Diagram Layout:
…
Required Entities:
sun image (I0)
…
"Sun" text label (T0)
…
Entity Relationships:
I1 is between I0 and I2
…
Entity Locations:
I0 is located at [8, 56, 13, 28]
…

**Input text:**
Caption:
[*insert new caption*]
Topic:
[*insert topic of caption*]

Figure 9: Input prompt given to the planner LLM during the initial diagram plan generation step.

**Instructions:**
Given a caption and a layout of a diagram, you should determine if something is wrong in the diagram based on the caption. You should explain your answer. Think step-by-step as to why the diagram is correct or not.
The diagram will be described in terms of entities in the diagram, the relationships between the entities, and the location of each entity.
An entity can be an image or text. Entity locations will be in [x, y, width, height] format, where 0,0 is the top left corner and 100,100 is max image size.

Here are some rules the diagrams should follow:
All numbers should be positive, no negative numbers.
Entities should not be outside the bounds [0, 0, 100, 100].
The x coordinate + the width of an entity should not exceed 100.
The y coordinate + the height of an entity should not exceed 100.
Entities of the same type should not overlap.

**In-context examples:**
[*diagram plans*]
What is wrong with this diagram?
"The moon phases are out of order, …"

**Input text:**
[*insert new diagram plan*]
What is wrong with this diagram?

Figure 10: Input prompt given to the auditor LLM during the diagram plan refinement step.

---

*Required Entities:*
egg image (I0)
larva image (I1)
pupa image (I2)
adult butterfly image (I3)
"egg" text label (T0)
"larva" text label (T1)
"pupa" text label (T2)
"adult butterfly" text label (T3)
*Entity Relationships:*
I3 has an arrow to I0
T3 labels I3
I1 has an arrow to I2
T0 labels I0
I0 has an arrow to I1
T2 labels I2
I2 has an arrow to I3
T1 labels I1
*Entity Locations:*
I0 is located at [24, 50, 14, 14]
I1 is located at [50, 74, 14, 14]
I2 is located at [74, 50, 14, 14]
I3 is located at [50, 24, 14, 14]
T0 is located at [20, 44, 10, 4]
T1 is located at [44, 80, 10, 4]
T2 is located at [80, 44, 10, 4]
T3 is located at [44, 20, 10, 4]

---

Figure 11: Example diagram plan from our LLM planner for the prompt "A diagram showing the life cycle of a butterfly, going from an egg to larva to pupa to an adult butterfly and repeating."

the diagram question-answering task rather than diagram generation, the diagram titles are often too short and don't provide enough information to produce meaningful diagram plans. Additionally, the dataset doesn't include descriptions for each object (*i.e.*, each object bounding box is labeled simply as 'blob'). To generate captions for each AI2D diagram, we present the diagram to LLaVA 1.5 and prompt it with the question "What is this diagram showing?". To collect region descriptions of the bounding boxes in AI2D, we first overlay the bounding box annotations of each object on the diagram, by assigning each box a label (*i.e.*, box 1 would get label "B1", box 2 would get label "B2", *etc.*.). Then we provide this annotated image to LLaVA 1.5 and ask the model to describe each box's content (*e.g.*, "what is the object labeled by 'B1'"?). An example of this annotation is shown in Fig. 4.

To ensure the quality of AI2D-Caption annotations, we perform a human evaluation of 50 LLaVA 1.5 annotations. For captioning, LLaVA generates very good captions 80% of the time, and for labeling, LLaVA generates very good bounding box region descriptions 68% of the time. We find that when LLaVA makes a captioning error, it is usually minor points, and for bounding box region errors, sometimes it may give nearby boxes the same description. While automatic annotations have some errors, we find that they are good enough for the domain adaption of DiagramGLIGEN; However, for the test set of AI2D-Caption and in-context learning annotations for the planner LLM, we manually annotate them to ensure correctness (Sec. 4.1).

## C   Additional DiagramGLIGEN Details

**Training.**   The parameters of DiagramGLIGEN are initialized from the GLIGEN (Box+Text checkpoint)[3], and trained on the AI2D-Caption dataset (see main paper Sec. 4.1 and Ap-

---

[3]https://github.com/gligen/GLIGEN

pendix B for details) for 15k steps (batch size of 5 per GPU), which takes 12 hours with 8 A6000 GPUs (each 48GB memory). Other hyperparameters include:

- Optimizer: AdamW (Loshchilov & Hutter, 2019)
- Learning Rate: 5e-5
- Warmup Steps: 2500
- Image Size: 512x512

**GLIGEN gated self-attention layers.** The high-level intuition of the gated self-attention layer is similar to cross-attention layers, which are used to incorporate extra information (*e.g.*, text, layouts) into the model. The cross-attention layer in the image generation backbone is used to incorporate text tokens for text-guided image generation, while the gated self-attention layer takes the grounding tokens for layout-guided image generation. Additional details are clarified in the GLIGEN paper (Li et al., 2023b).

## D  Experimental Setups

### D.1  Metrics

**VPEval (Objects, Counts, Relationships, Texts).** We evaluate the diagrams in terms of the presence of objects (object evaluation), the number of objects (count evaluation), the correctness of spatial and connection relationships (relationship evaluation), and the presence of correct text labels (text evaluation) using the VPEval metric (Cho et al., 2023b). VPEval works by first generating evaluation programs that call specific evaluation modules (*e.g.*, Object, OCR, VQA) and then running the modules to evaluate the image. Each module evaluates different parts of the image (*e.g.*, Object checks object presence). For object, count, and text rendering evaluation, we use the ground-truth diagram plan to determine which evaluation programs to use. For relationship evaluation, following VPEval, we use an LLM (GPT-4) to generate VQA questions that evaluate the spatial/connection relationships of objects in the diagrams (*e.g.*, if the moon is in between the sun and earth or if a light bulb is connected to a battery). While the original VPEval uses BLIP-2 (Li et al., 2023a) as its visual reasoning model, we employ the recently released LLaVA 1.5 (Liu et al., 2023a) model as our visual reasoning model, as we find it is more faithful in diagram question answering in our initial experiments and is an overall stronger model than BLIP-2 (Liu et al., 2023a). In our initial experimentation, we found that LLaVA 1.5 sometimes struggles with relationship evaluation, so we fine-tune the model on relationships to ensure accurate evaluation.

For object evaluation, LLaVA determines if the object is present. For count evaluation (*i.e.*, if a diagram requires multiple instances of an object), we ask LLaVA if there are exactly $N$ instances of the object in the diagram, where $N$ represents the count of that object in the ground-truth diagram. For relationship evaluation, we ask LLaVA if the relation (spatial or connection) is true. In our experiments, we find that LLaVA often can generate false positives during relation evaluation, as such we also show human evaluation for object relationships (main paper Sec. 4.4). For text evaluation, in alignment with VPEval, we utilize EasyOCR (AI, 2023) as the OCR model and check if the target text is detected.

## E  Human Evaluation Setup Details

We employ crowd-workers from Amazon Mechanical Turk (AMT) for our human preference study. To ensure high-quality annotations, we set the following requirements for the workers: they must possess an AMT Masters qualification, have completed more than 1000 HITs, maintain an approval rating above 95%, and come from the United States, Great Britain, Australia, or Canada, given that our task is in English. We pay workers $0.06 to compare two diagrams (roughly $14-15/hr). For each prompt, we show diagrams generated by both our DiagrammerGPT and the fine-tuned Stable Diffusion v1.4 (the order of diagrams are randomly shuffled every time to prevent selection biases) and ask five annotators to indicate their preference based on (1) the accuracy of the generated relationships between objects

Figure 12: Interface provided to annotators for human evaluation.

(*e.g.*, spatial relationships and arrows/lines) and (2) alignment to the input prompt (*e.g.*, how well does the generated diagram reflect the input prompt). Then, we take the agreement of the annotators. The task is described to the annotators as such:

1. Object Relationships is a measure of which diagram better captures the proper relationships of the objects (*i.e.*, spacing/positioning of them, arrows/lines between them, *etc*.).

2. Alignment is a measure of which diagram is a better representation of the input sentence.

Fig. 12 shows the interface provided to the annotators during the human evaluation process.

## F  Additional Experiment Results

### F.1  Qualitative Results and Analysis

**Qualitative comparison to baselines.**  Fig. 13 shows example diagrams generated by the baselines (Stable Diffusion v1.4 and AutomaTikZ) and our DiagrammerGPT (both diagram plan and final generation diagram) on the AI2D-Caption test split. Our diagram plans strongly reflect the prompts and the final diagrams are more aligned to the input prompts. In Fig. 13 top example, our diagram correctly shows the earth in four phases revolving around the sun and in the second example, our diagram plan correctly represents the life cycle of a butterfly and the generated diagram captures the circular flow of the diagram plan as well most aspects of the life cycle. Stable Diffusion v1.4 either over- or under-generates objects in the image (*e.g.*, too many earths in the first example and missing egg/larva/pupa stages in the second example), and AutomaTikZ fails to generate proper layouts and objects. Although our generated diagram plans are generally correct, however, sometimes DiagramGLIGEN can fail to properly follow all aspects (*e.g.*, the egg is misdrawn and the larva/pupa are swapped in Fig. 13 bottom example). As noted in main paper

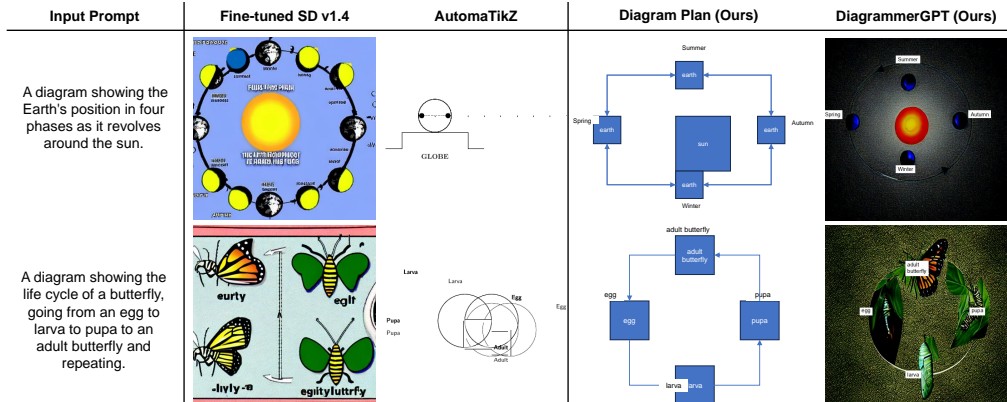

Figure 13: Example diagram generation results from baselines (fine-tuned Stable Diffusion v1.4 and AutomaTikZ) and our DiagrammerGPT on the AI2D-Caption test split. In the first example, our DiagrammerGPT correctly gets the object count right and has clear text, whereas Stable Diffusion v1.4 overpopulates the entities orbiting around the sun. In the second example, our DiagrammerGPT generates an accurate diagram plan and a diagram that mostly reflects the plan, whereas Stable Diffusion v1.4 fails to show a life cycle (*i.e.*, missing the egg, pupa, and larva). As noted in main paper Sec. 5.2, once a better backbone becomes available, our DiagrammerGPT can produce better diagrams based on the diagram plans. AutomaTikZ struggles to generate the proper layouts and objects for both examples.

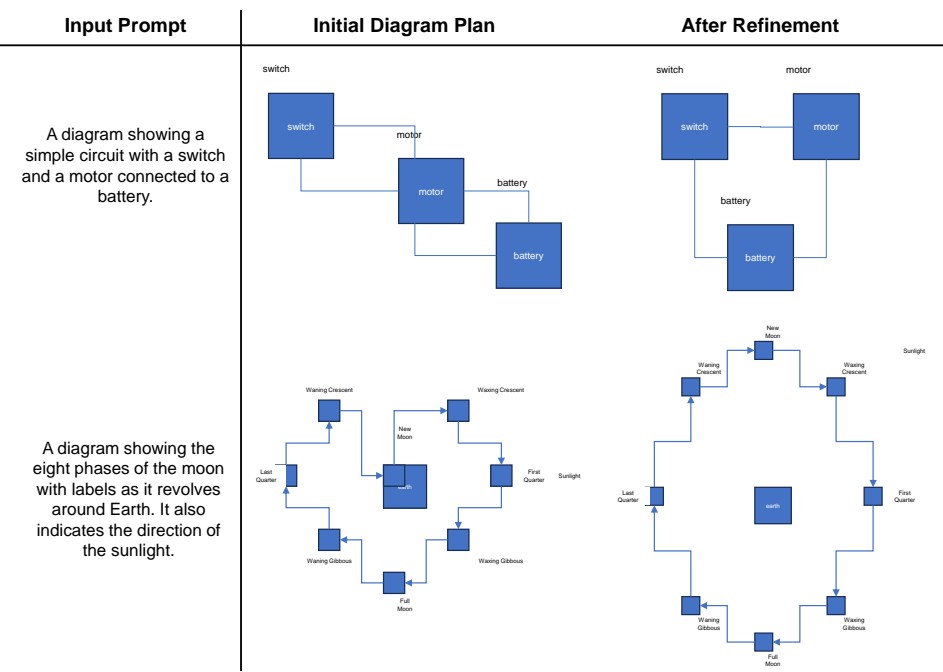

Figure 14: Examples from our diagram refinement step. Our auditor LLM can help reorganize the connections between the components to be more clear in the first example and prevent overlaps of objects in the second example.

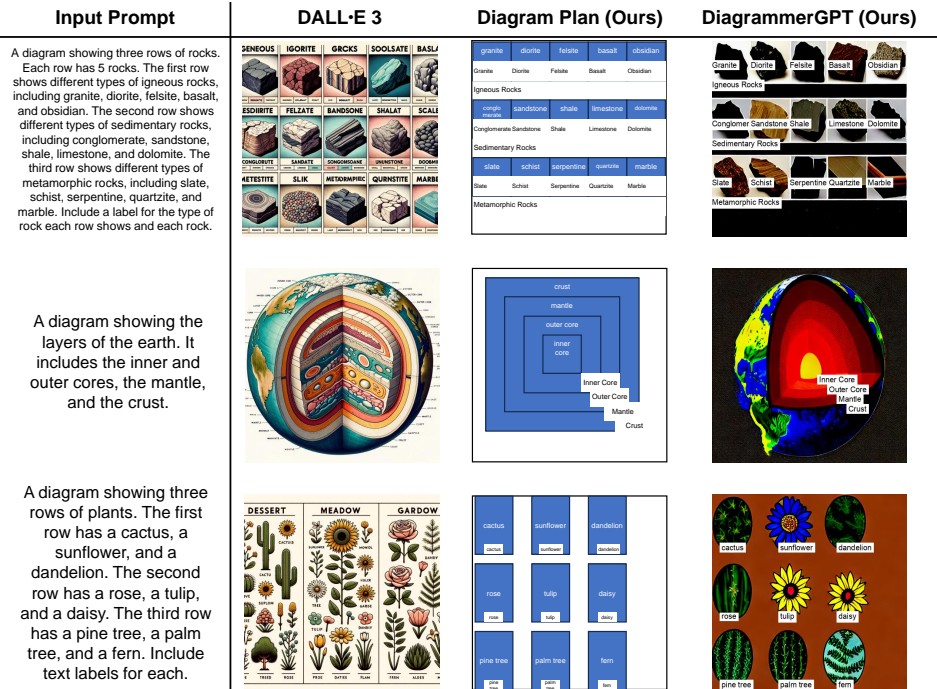

Figure 15: Examples of open-domain generation demonstrate that our DiagrammerGPT can create diagrams that adhere to the input text prompts. Although DALL-E 3 yields images with superior visual quality, it tends to generate diagrams with redundant and crowded objects and also struggles to follow the prompt accurately (*e.g.*, in the second example, it is not clear where the locations of layers such as the 'inner core', 'outer core', and 'mantle' are. In the third example, it generates too many objects that are not in rows).

Sec. 5.2, once a better backbone becomes available, our DiagramGLIGEN can produce better diagrams following the diagram plans.

**Diagram plan refinement.** In Fig. 14, we show how our diagram refinement step (see main paper Sec. 3.1) improves the diagram plans. In the top example, the switch is not connected to the battery, thus does not affect the circuit. After refinement, the connections are corrected so the switch is now also connected to the circuit and the layouts are adjusted to have a more straightforward flow. In the bottom example, the moon phase of 'New Moon' is too low and overlaps with the 'Earth' object. After refinement, there is no more overlap.

F.2   Additional Analysis

**Open-domain diagram generation.** Our main diagram generation experiments are conducted on diverse domains such as astronomy, biology, and engineering which are included in the LLM planner's in-context examples. However, given that the in-context examples do not encompass all diagram domains, we experiment with generating diagrams in areas not covered by our LLM in-context examples, such as geology and botany, to assess whether our DiagrammerGPT maintains its ability to produce more accurate diagrams in previously unseen domains.

In Fig. 15, we show examples of comparing our open-domain diagram generation to DALL-E 3. While our DiagramGLIGEN struggles in some cases (like the third example), it is able to strongly adhere to the diagram plan. Fig. 16 (bottom) also shows our LLM planner is easily able to generalize to completely new domains (*e.g.*, neural networks and vacation planning). As mentioned in main paper Sec. 5.2, once a stronger layout-guided image generation model than GLIGEN with Stable Diffusion v1.4 backbone is available, our DiagrammerGPT

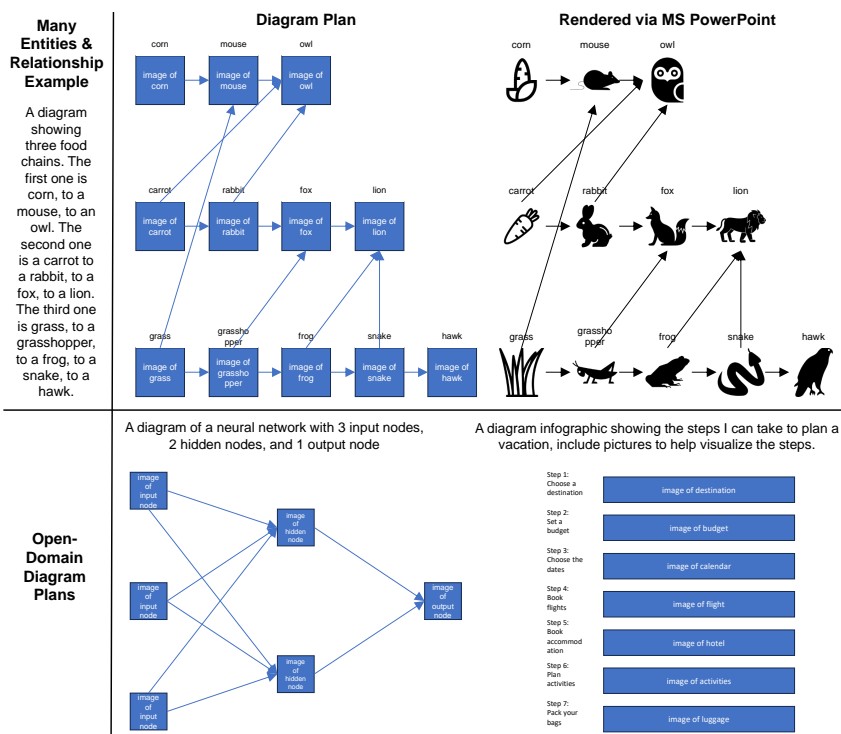

Figure 16: Additional examples. Our planner LLM is effective at generating dense (top) and open-domain (bottom) diagram plans.

can produce higher quality results. We find that DALL-E 3 generally produces images with good aesthetic style but tends to generate diagrams with redundant and crowded objects (*e.g.*, excessive unnecessary text descriptions in the rock and Earth examples, and an overabundance of plants in the third example). It also continues to struggle with creating accurate diagrams that adhere to a prompt (*e.g.*, generating incorrect layers in the earth example and generating three columns of plants instead of three rows in the plant example). The DALL-E 3 system card (OpenAI, 2023a) also notes that DALL-E 3 tends to generate scientifically inaccurate information in diagrams.

**Vector graphic diagram generation in different platforms.** We render our diagram plans in Microsoft PowerPoint via VBA language,[4] Inkscape[5] via a Python scripting extension[6], and Adobe Illustrator[7] via JavaScript. We represent objects using icons, which are retrieved via the Noun Project Icons API based on corresponding text descriptions.[8] Fig. 16 (top) and Fig. 18 show additional examples of diagram plans rendered in the other platforms.

**GPT-4 *vs.* GPT-4Vision for diagram plan creation and refinement.** As described in main paper, our DiagrammerGPT employs a text-only GPT-4 model for diagram planning and refinement. To explore whether a multimodal language model can offer improvements over text-only GPT-4, we experiment with using the recently introduced GPT-4Vision (GPT-V) model (OpenAI, 2023c) as the *planner* and *auditor* LLM during the diagram generation and refinements steps. As the GPT-4V model does not provide API access yet, we conduct a small-scale qualitative study via the ChatGPT web UI. In our experiments, for the diagram plan creation stage (see main paper Sec. 3.1), GPT-4V does not provide improvements over

---

[4]https://learn.microsoft.com/en-us/office/vba/api/overview/powerpoint

[5]https://inkscape.org

[6]https://github.com/spakin/SimpInkScr

[7]https://www.adobe.com/products/illustrator.html

[8]https://thenounproject.com/api/

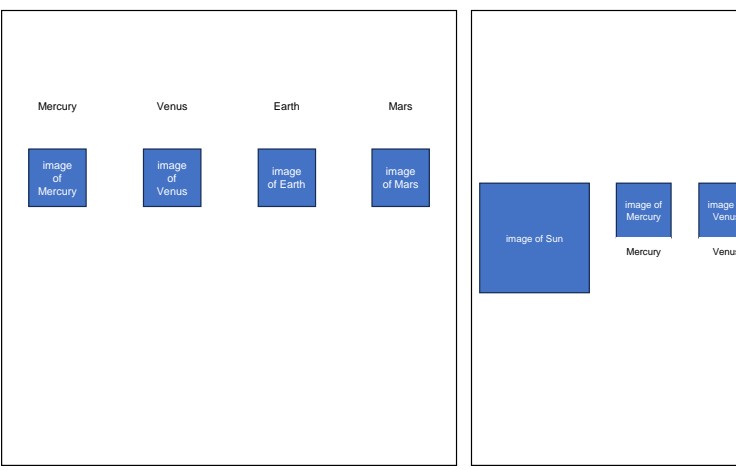

Figure 17: Example of two GPT-4 generated diagram plans. Given the same prompt, GPT-4 can generate diverse plans between runs.

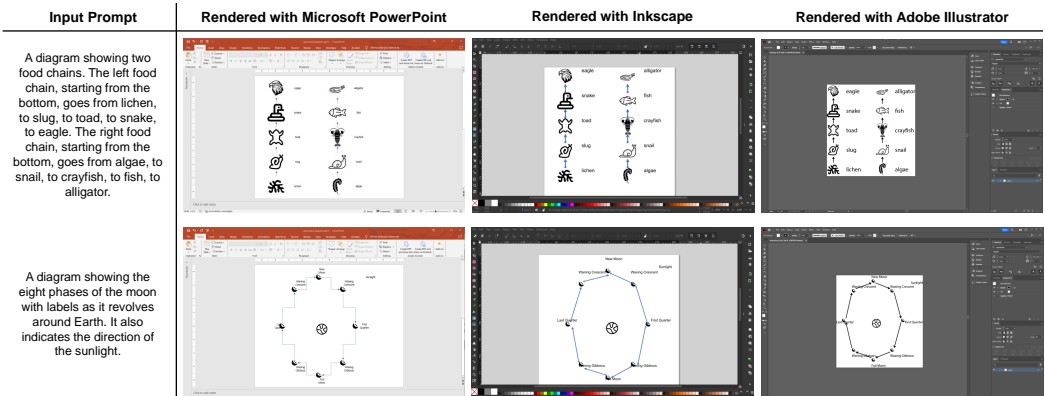

Figure 18: Examples of vector graphic diagrams generated with our diagram plans and exported into Microsoft PowerPoint, Inkscape, and Adobe Illustrator.

text-only GPT-4. In Fig. 19, we present a comparison between the diagram plans generated by GPT-4 and GPT-4V. GPT-4V does not produce diagram plans that are better than text-only GPT-4, suggesting that our text-only representation is robust enough until better or fine-tuned versions of GPT-4V become available for diagrams. Similarly, during the diagram refinement step (see main paper Sec. 3.1), we observed that GPT-4V tends to overestimate correctness when compared to text-only GPT-4, further indicating the strength of our text representation. Fig. 20 shows two examples comparing the models. While text-only GPT-4 is not perfect, it can identify some errors, whereas GPT-4V says the diagram does not need improvement.

**Using model based text rendering instead of Pillow.** We also experiment with using a model-based text renderer, TextDiffuser-2 (Chen et al., 2023b) instead of Pillow. Fig. 21 shows that while TextDiffuser-2 is capable of producing good text labels, however, it can sometimes merge letters (*e.g.*, the "mm" in summer). Pillow guarantees there is no rendering error (and can easily allow font color/size adjustments). Due to the modular nature of

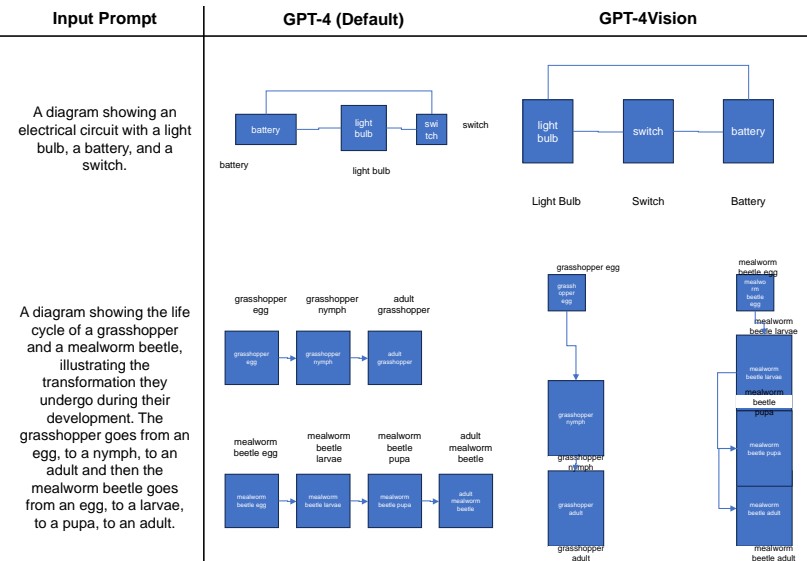

Figure 19: Comparison examples of diagram plans generated by GPT-4 and GPT-4Vision (GPT-4V). GPT-4 creates diagram plans that are sufficiently accurate in capturing the presence of objects and their relationships and GPT-4V does not provide plans that are better.

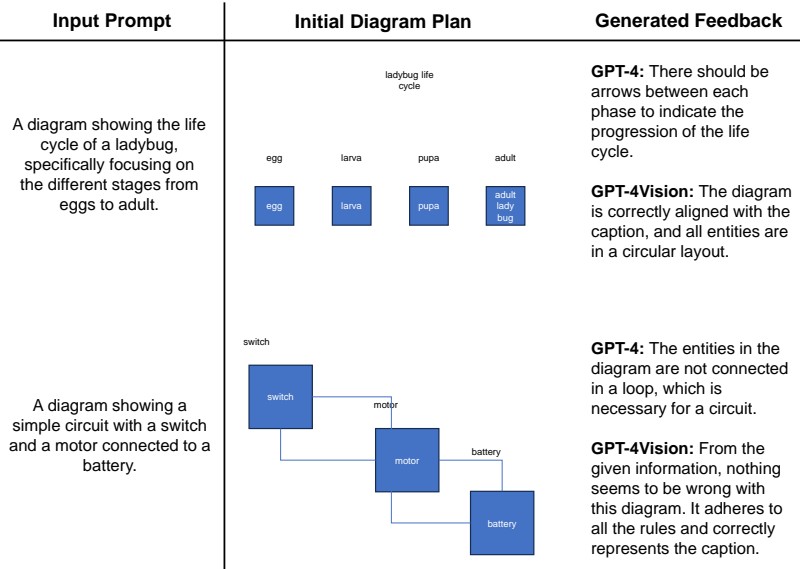

Figure 20: Comparison examples of GPT-4 *vs.* GPT-4Vision for diagram plan refinement. While text-only GPT-4 is not perfect, it can identify the errors, whereas GPT-4Vision says the diagram does not need improvement.

DiagrammerGPT, using a text rendering model can easily be incorporated if the end user wants.

**Is the LLM capable of generating diverse diagram plans?**   We find that having an LLM generate the diagram plans results in a wide diversity of diagrams. As we show in Fig. 13, 14, 15, and 16, the LLM can generate plans for many different prompts and domains. In Fig. 17 we show an example of the LLM generating variations of the same prompt, further indicating the LLM (GPT-4 in our case) is capable of producing diverse diagram plans.

**Our Explicit Text Rendering**      **Text Rending via TextDiffuser-2**

Figure 21: Explicit text rendering via Pillow compared to model-based text rendering via TextDiffuser-2 (Chen et al., 2023b).

## F.3 Ablation Studies

We show ablation studies on our design choices with DiagramGLIGEN: the number of denoising steps with layout guidance and whether to only update the gated self-attention layers parameters.

| # Layout guidance steps | VPEval | | | |
|---|---|---|---|---|
| | Object | Count | Text | Relationships |
| $\alpha = 5$ steps | **88.0** | 50.6 | **48.0** | 84.9 |
| $\alpha = 10$ steps | 86.4 | 49.4 | 47.0 | 86.6 |
| $\alpha = 15$ steps (default) | 86.4 | **57.0** | 47.5 | **87.9** |

Table 5: Ablation of # denoising steps with layout guidance. DiagramGLIGEN uses 50 denoising steps in total. We use $\alpha = 15$ steps as our default setting.

**Number of denoising steps with layout guidance.** The number of denoising steps with layout guidance (*i.e.*, with the gated self-attention layer activated in each transformer block of the diffusion UNet), denoted as $\alpha$, is a crucial hyper-parameter in DiagramGLIGEN. A larger $\alpha$ value indicates stronger layout control. Table 5 presents an ablation study using varying $\alpha$ values. A smaller $\alpha$ value enhances object generation, while a larger value improves count performance. This observation aligns with intuition: rigorous layout control more effectively prevents the generation of extraneous objects in the background but may detract from the visual realism of the generated objects, which is also observed in (Li et al., 2023b; Lin et al., 2023). We set the default value for $\alpha$ as 10 steps, as it ensures a good balance of accuracy for objects and counts while achieving optimal performance in depicting object relationships.

| Updated parameters | VPEval | Captioning | CLIPScore |
|---|---|---|---|
| | Overall | BERTScore | Img-Txt |
| None | 68.5 | 88.9 | 29.3 |
| GatedSA Layers only | 67.4 | 88.9 | 30.0 |
| All Layers (default) | **71.2** | **89.4** | **32.1** |

Table 6: Ablation of fine-tuning different layers of DiagramGLIGEN. We use the fully fine-tuned model as our default setting. *GatedSA: Gated Self-Attention.*

| Diagram Plan Source | VPEval | Captioning | CLIPScore |
|---|---|---|---|
| | Overall | BERTScore | Img-Txt |
| Ground-truth + DiagramGLIGEN (oracle) | **81.9** | **89.8** | **32.3** |
| GPT-4 + DiagramGLIGEN | 71.2 | 89.4 | 32.1 |

Table 7: Ablation of using ground-truth plans from AI2D-Caption (*e.g.*, oracle performance) instead of GPT-4.

**Fine-tuning: all layers vs. layout layers.** In Table 6, we present an ablation study comparing the fine-tuning of only the layout control layers in DiagramGLIGEN with fine-tuning of the entire model, including the Stable Diffusion backbone. Full fine-tuning enhances performance and improves the visual quality of the diagrams. Therefore, we employ the fully fine-tuned version as our default model for all subsequent experiments.

**Using ground-truth diagram plans.** We experiment with generating diagrams using ground-truth diagram plans from AI2D-Caption. Doing this allows us to measure the upper bound of DiagramGLIGEN and see how much room our stage 1 has for improvement. Table 7 shows that using ground-truth plans does indeed do better than GPT-4 and that DiagramGLIGEN is able to perform better when using ground-truth plans. However, it is interesting to know that using GPT-4 performs very closely to the oracle score.

# G   Limitations

Our framework can benefit many educational applications, such as presentation/paper creation, and human-in-the-loop diagram generation/modification. However, akin to other text-to-diagram/text-to-image generation frameworks, our framework can also make some errors and be utilized for potentially harmful purposes (*e.g.*, creating false information or misleading diagrams), and thus should be used with caution in real-world applications (with human supervision, *e.g.*, as described in Sec. 5.4 human-in-the-loop diagram plan editing). Also, generating a diagram plan using the strongest LLM APIs can be costly, similar to other recent LLM-based frameworks. We hope that advances in quantization/distillation and open-source models will continue to lower the inference cost of LLMs. Lastly, DiagramGLIGEN is based on the pretrained weights of GLIGEN and Stable Diffusion v1.4. Therefore, we face similar limitations to these models, including deviations related to the distribution of training datasets, imperfect generation quality, and only understanding the English corpus.

