# OpenReview forum: "DiagrammerGPT: Generating Open-Domain, Open-Platform Diagrams via LLM Planning"
_colmweb.org/COLM/2024/Conference — COLM_

### Official Review · Reviewer_Bhqq · 2024-05-11

**Rating:** 6
**Confidence:** 3
**Ethics Flag:** 1

**Summary:**

This paper introduces DiagrammerGPT, a novel framework for the challenging task of diagram generation. The authors developed AI2D-Caption, a new dataset to benchmark text-to-diagram generation. DiagrammerGPT employs a two-stage process: the first stage uses LLMs to create and refine 'diagram plans' through a planner-auditor feedback loop, while the second stage utilizes DiagramGLIGEN and a text label rendering module to generate diagrams based on these plans. The framework also supports features such as open-domain generation, vector graphic creation, human-in-the-loop editing, and multimodal planning.

**Questions To Authors:**

Many questions are raised in the weakness section, please refer above.

(1) In-Context Learning Details: The authors mentioned annotating 30 diagrams for in-context learning. How many diagrams were used to prompt each example? How were the supportive examples for ICL selected—randomly or using similarity functions?

(2) Figure Clarity (Comment): This won't affect my rating, but some figures have text that is difficult to read due to its small size. It would be beneficial to enlarge the figures or at least the text labels for better readability.

**Reasons To Accept:**

(1) **Novelty and Relevance**:
This paper addresses a unique and challenging problem in diagram generation, presenting a fresh perspective on a multimodal task that encompasses diverse challenges such as commonsense reasoning, layout planning, and image generation.

(2) **Initial Contributions**:
As a pioneering work in a new research area, this paper conducts foundational experiments demonstrating the effectiveness of the proposed method, including LLM planning with iterative feedback and the DiagramGLIGEN module.

(3) **Clarity and Structure**:
The manuscript is well-structured and clear, with effective use of illustrations that enhance the understanding of core concepts and methodologies, making the proposed method more accessible to readers.

**Reasons To Reject:**

(1) **Lack of Ablation Studies:** The paper does not include sufficient ablation studies to analyze the various factors influencing the performance of this complex multimodal task, making it difficult to fully understand the strengths and limitations of the proposed pipeline (further details are provided in subsequent points).

(2) **Lack of Baseline Comparison:** Table 1 presents results based on constraints generated by the LLM in stage 1, but lacks a baseline comparison using ground truth diagram planning annotations. Including this baseline would clarify the T2I performance with accurate diagram plans.

(3) **Limited Discussion on Diagram Planning:** The paper discusses the performance of DiagramGLIGEN in stage 2 but lacks a thorough exploration of the diagram planning in stage 1. Specifically, there is no discussion of related work in the planning stage, and only GPT-4 results are presented. It would be beneficial to evaluate other LLMs for this task and consider if state-of-the-art models in procedure planning could be applicable to diagram planning.

---

> ### Author Rebuttal · Authors · 2024-05-31
>
> Thank you for your valuable feedback. Below we address your concerns with further clarifications & experiments.
>
> **W1. Ablation studies:**
> We would like to bring your attention to Appendix F.3, where we present ablation studies for different number of denoising steps, choosing which layers to fine-tuning for DiagramGLIGEN, and different LLMs for planning.
>
> **W2. Comparison to diagram generation from GT layouts:**
> Following your suggestion, we generate diagrams from GT diagram plans and present the results here:
>
> |  |  |  |  |
> |-|:-:|:-:|:-:|
> |  | VPEval Overall | BERTScore | CLIP Img-Txt |
> | GPT-4 + DiagramGLIGEN | 71.2 | 89.4 | 32.1 |
> | GT Plans + DiagramGLIGEN (Oracle) | 81.9 | 89.8 | 32.3 |
>
>
> From this, we see that using GT diagram plans leads to higher overall performance, however, using diagram plans from our first stage is not too far behind. This is also reflected in Table 2 where we show that humans label GPT-4 diagram plans very highly (4.96 / 5 object presence and 4.72 / 5 object relations).
> Thanks for the suggestion, and we will add the results to the paper.
>
> **W3. Different LLMs:**
> In Appendix F.3, we present layout generation experiment results from GPT-3.5 as well. As mentioned in the section, we also tried LLaMA2-Chat (13B) via both zero-shot and fine-tuning, but we found it does not perform well, often missing objects or failing to generate meaningful plans.
>
> We also just tried the recently released LLaMA3 (8B) model but even with fine-tuning, it fails to produce good diagram plans, often just repeating the same object, missing objects, or sometimes failing to produce meaningful plans. We summarize the result below:
>
>
> |  |  | Layout Recall |  |
> |-|:-:|:-:|:-:|
> |  | Object | Text | Overall |
> | LLaMA3 8B | 24.9 | 32.2 | 29.2 |
> | GPT-3.5 Turbo | 82.5 | 54.6 | 74.7 |
> | GPT-4 (default) | **84.1** | **60.1** | **78.4** |
>
>
> Layout recall checks whether certain objects/texts present in GT diagram plans are present in the generated diagram plan.
>
> The LLaMA models may potentially be suffering from the limited training data.
>
> **Q1. In-context details:**
> For each of the three domains (astronomy, biology, engineering), we use 10 in-context examples for each prompt. For open-domain prompts, we use all 30 in-context examples. The examples chosen for in-context use were randomly selected from the AI2D dataset.
>
> **Q2. Figure enlargement:**
> We will try to increase the text label/figure size where needed to improve readability. Thanks for the suggestion.

---

> ### Author Response · Authors · 2024-06-04
> **A Gentle Reminder for Response**
>
> We thank the reviewer for their time and effort in reviewing our paper.
>
> We hope that our response has addressed all the questions and hope that the reviewer can consider revising the score based on our response. We are also happy to discuss any additional questions.
>
> With sincere regards,
> The authors

---

> > ### Comment · Reviewer_Bhqq · 2024-06-04
> > **Response to Rebuttal**
> >
> > I have read the authors' rebuttal response, and found most of my questions answered. I will remain the same positive rating.

---

### Official Review · Reviewer_SF1t · 2024-05-12

**Rating:** 6
**Confidence:** 3
**Ethics Flag:** 1

**Summary:**

This paper presents a text-to-diagram approach named DiagrammerGPT. It uses a two-step process to create open-domain diagrams from textual prompts, where GPT4 is prompted to generate a diagram plan followed by a visual generator creating the actual diagram from the generated diagram plan.

Initially, the authors use GPT-4 to create diagram plans from text prompts during the diagram planning stage, which includes listing entities, their relationships, and layouts. Afterward, leveraging recent advances, another Language Model acts as an auditor to identify errors such as misplaced objects. The plan is then refined based on this feedback to correct inaccuracies like incorrect sizes, resulting in notably accurate LLM-generated diagram plans with minimal errors.

During stage two, a new model called DiagramGLIGEN is designed for creating diagrams from plans and clearly rendering text labels for legibility. It's built upon the GLIGEN framework and trained using the AI2D-Caption dataset, which includes gated self-attention layers in the Stable Diffusion v1.4 model. To overcome the difficulty of text rendering in existing T2I models, the authors directly overlay text on diagrams to ensure clarity.

Both human evaluation and LLaVA/ClIP based evaluation show that the proposed approach can generate diagrams with better quality.

**Questions To Authors:**

As I mentioned above, a lot of important details are missing. Besides the forementioned in-context examples, here are a list of points need to be further clarified:
- The description of the model in Section 3.2 is somehow vague. I suggest the authors can provide a detailed model architecture diagram about GLIGEN and how the new gated self-attention layer is attached. The authors need to explain more about the intuition behind adding the gated self-attention layer, as this is an important model contribution.
- AI2D-Caption is an important part of the paper. The authors state that they select 75 diagrams. What are the selection criteria?
- Human evaluation for pairwise preference is mentioned at Section 4.4. Please describe how the labelling task is described to annotators to avoid potential bias, as this is important for the faithfulness of the result.
- For the vector graphic diagram generation, it is not clear how this is implemented. The current version is not replicable.

**Reasons To Accept:**

1. Text-to-diagram is a relatively new topic, and it is very useful in practice.
2. The proposed two-stage approach is technically reasonable.

**Reasons To Reject:**

Some important details are missing. For example, the quality of generated plan largely depends on the in-context examples. However, the authors do not introduce how to pick up a set of in-context examples in an open-domain scenario. Are the set of in-context examples selected on-the-fly or it is a static set?

---

> ### Author Rebuttal · Authors · 2024-05-31
>
> Thank you for your valuable feedback. Below we address your concerns with further clarifications.
>
> **W1. In-context details:**
> For open-domain diagram generation (Sec. 5.4 / Fig. 5 &14), we use all 30 in-context examples mentioned in Sec. 4.1. These examples do not change (i.e. static). We find that the LLM (GPT-4) can generalize these examples into other domains well (see Fig. 5, 14, & 15). We will clarify this in the paper.
>
> **Q1. GLIGEN details:**
> The high-level intuition of the gated self-attention layer is similar to cross-attention layers, which are used to incorporate extra information (e.g., text, layouts) into the model. The cross-attention layer in the image generation backbone is used to incorporate text tokens for text-guided image generation, while the gated self-attention layer takes the grounding tokens for layout-guided image generation.
>
> Additional details are clarified in the GLIGEN paper (Li et al., 2023b). However, following your suggestion, we can clarify the important architectural details in our paper.
>
> **Q2. AI2D-Caption details:**
> For the test set of AI2D-Caption, we randomly choose 25 diagrams from astronomy (e.g., solar system, etc.), biology (e.g., life cycles, etc.), and engineering (e.g., circuits, etc.) for a total of 75 diagrams. We will add the selection criteria to the paper.
>
> **Q3. Human evaluation instructions:**
> Here is how we describe the image-text alignment and object relationships to annotators:
>
> - Object Relationships is a measure of which diagram better captures the proper relationships of the objects (i.e. spacing/positioning of them, arrows/lines between them, etc.).
> - Alignment is a measure of which diagram is a better representation of the input sentence.
>
> The annotators are then shown a diagram from DiagrammerGPT and SD v1.4 and provided 3 options: Diagram A, Diagram B, and Equal. The diagrams are _randomly shuffled_ so there is no pattern between which model produces diagram A or B.
>
> We will add a screenshot of the setup to our paper to make sure these details are clarified.
>
> **Q4. Vector graphic generation implementation:**
> We take the diagram plan generated by the LLM and convert it into a format that is accepted by the related vector graphic platform (Appendix F.2; e.g., VBA for PowerPoint) via a script. Objects are represented using icons automatically pulled from the public icon library “The Noun Project” (Appendix F.2). We will release all related code for render vector graphic rendering.

---

> > ### Comment · Reviewer_SF1t · 2024-06-04
> > **To the rebuttal**
> >
> > Thanks for the response! It addresses most of my questions.
> > I remain a positive score to this paper. As a brand-new application of generative language model, I think this paper may have value to readers.

---

> ### Author Response · Authors · 2024-06-04
> **A Gentle Reminder for Response**
>
> We thank the reviewer for their time and effort in reviewing our paper.
>
> We hope that our response has addressed all the questions and hope that the reviewer can consider revising the score based on our response. We are also happy to discuss any additional questions.
>
> With sincere regards,
> The authors

---

### Official Review · Reviewer_GeX9 · 2024-05-20

**Rating:** 7
**Confidence:** 4
**Ethics Flag:** 1

**Summary:**

The paper proposes DiagrammerGPT, a method to generate diagram images using a combination of LLMs and diffusion models. The method first generates layouts of the objects, including bounding boxes, paired text annotations, and arrows or lines between objects. This stage involves in-context learning and self-correction loops to refine the layouts. Then a GLIGEN-based layout-to-image model is fine-tuned and used for generating the diagram image based on the layouts. A new dataset named AI2D-Caption is generated based on an existing diagram dataset for the task and experiment. Experimental results show a clear improvement of the method over existing baselines.

**Questions To Authors:**

- Does the AI2D dataset originally include arrows/lines between objects? If not, how did you get the annotation for the dataset?
- Could you clarify the difference between fine-tuned GLIGEN in VPGen and the DiagrammarGLIGEN? For input, GLIGEN is like box+text while DiagrmmarGLIGEN is box+text+relation? Is GLIGEN also fine-tuned on all layers like in Table 5?

**Reasons To Accept:**

- The problem of diagram creation as image generation is interesting and has practical values. Treating it as an image generation has some advantages over program-based rendering like image diversity, flexible control, and having open-domain objects.
- The proposed method, while similar to existing ones, involves arrows or lines for object relation in the first stage, which introduces more fine-grained elements to the layouts. The method also largely outperforms existing baselines, showing its effectiveness. The proposed dataset is also useful for future work.

**Reasons To Reject:**

- Technical novelty is limited. There are some new elements like the use of arrows/lines or the iterative refinement process in using LLMs to generate layouts. But most of the techniques are not new like the in-context learning design or DiagramGLIGEN (basically a simple variant of GLIGEN).
- Lack of in-depth experiment/analysis. While I appreciate the significant improvement shown in Table 1&2, it is not quite obvious where the improvement comes from. Why would the fine-tuned VPGen get decreased performance in count, CIDEr, and BERTScore compared to its zero-shot version? Besides, does the improvement of DiagrammerGPT over fine-tuned VPGen come from the first or second stage? I think it would be better to compare the first stage performance of fine-tuned VPGen and the proposed method.

---

> ### Author Rebuttal · Authors · 2024-05-31
>
> Thank you for your valuable feedback. Below we address your concerns with further clarifications and experiments.
>
> **W1. Paper contributions:**
> We would like to remind you of the strong contributions of our project:
>
> 1. We present a new text-to-diagram generation framework that leverages the knowledge of LLMs for planning & refining the open-domain diagrams (Sec. 3), which is more effective than existing T2I methods (Table 1 & 3; Fig. 4 & 5).
> 2. We show that iterative self-refinement can help create better diagram plans (Table 2; Fig. 13).
> 3. We show that our diagram plans can be rendered via any vector graphic software (Fig. 6, 15, & 16) as well as T2I models, allowing for more flexible real-world use (e.g., for teachers) and human-in-the-loop editing (Fig. 7).
> 4. We produce a new dataset (AI2D-Caption; Sec. 4.1) with evaluation suites (VPEval + LLaVA 1.5 finetuned on diagrams; Sec. 4.3) that can help the community benchmark and improve the text-to-diagram methods.
> 5. We train DiagramGLIGEN to have improved control of text labels/entity relations  (Sec. 3.1).
>
> **W2. VPGen additional analysis:**
> Our improvement over VPGen primarily comes from stage 1. Following your suggestion, we compare fine-tuned VPGen to our DiagrammerGPT in the first stage.
>
> |  |  | Layout Recall |  |
> |-|:-:|:-:|:-:|
> |  | Object | Text | Overall |
> | Fine-tuned VPGen | 30.6 | 0 | 12.9 |
> | DiagrammerGPT (ours) | **84.1** | **60.1** | **78.4** |
>
> Layout recall compares GT diagram plans to the generated diagram plan.
>
> As shown in the table, the fine-tuned VPGen struggles to produce the required entities for a diagram, while DiagrammerGPT does better.
>
> The reason the fine-tuned VPGen does worse sometimes compared zero-shot VPGen is likely due to the diagram planning stage. After fine-tuning, VPGen generates objects in the right domain (as reflected by the increased object score 55.8 $\rightarrow$ 62.8 in the main paper Table 1). However, potentially due to limited training data, it often generates too many objects, overlaps objects, etc. which leads to lower scores in certain metrics.
>
> **Q1. Arrow/line annotations:**
> The original AI2D dataset labels which objects are connected via a line/arrow.
>
> **Q2. Fine-tuned GLIGEN in VPGen vs. DiagramGLIGEN:**
> We fine-tune the GLIGEN model in VPGen like we do for DiagrammerGPT to give a fair comparison. DiagrammerGPT still has a different diagram planning stage than VPGen and also explicitly renders text labels, whereas VPGen does not.

---

> > ### Comment · Reviewer_GeX9 · 2024-06-06
> > **Thank you for your response**
> >
> > Most of my concerns are addressed in the response. I hope the authors could incorporate these results and clarifications into the next version of the paper. I have raised my score accordingly.

---

> ### Author Response · Authors · 2024-06-04
> **A Gentle Reminder for Response**
>
> We thank the reviewer for their time and effort in reviewing our paper.
>
> We hope that our response has addressed all the questions and hope that the reviewer can consider revising the score based on our response. We are also happy to discuss any additional questions.
>
> With sincere regards,
> The authors

---

### Official Review · Reviewer_Lutp · 2024-05-24

**Rating:** 6
**Confidence:** 5
**Ethics Flag:** 1

**Summary:**

This paper presents DiagrammerGPT, which leverages large language models (LLMs) to produce text-visual diagrams. With the powerful capability of layout planning, they first derive and refine a diagram plan via LLMs. Then, they can follow the plan to generate the resulting diagram via the layout-grounded diffusion model.

**Questions To Authors:**

Please see Reasons To Reject

**Reasons To Accept:**

- This paper is well-written and easy to follow.
- The targeted text-to-diagram is a novel task, which is a crucial application of the general text-to-image synthesis (T2I).
- The proposed iterative refinement is well-motivated, and the grounded diffusion can also facilitate the output quality.
- They provide comprehensive human evaluations as well as qualitative examples.

**Reasons To Reject:**

- Since there are already previous works [1,2,3] adopting LLMs to improve layout planning, the novelty of this paper can be an issue. In addition, the proposed DiagramGLIGEN is an extension of the existing GLIGEN [4].
- Although the iterative feedback seems to be helpful, there should be an empirical study to show the effectiveness of this refinement.
- One powerful feature of current T2I is the diversity generation. Can LLMs also produce diverse yet meaningful diagram plans? And can those layouts further generate different visual results? Will this text-to-layout technique, which helps better controllability, hurt the diversity of T2I?
- From those presented visualized cases, they look not that visually appealing for actual human usage (correct me if this is not true). I am worried about the real-world practicality with the current pipeline.

**Reference**
- [1] LayoutGPT: Compositional Visual Planning and Generation with Large Language Models
- [2] LLM-grounded Diffusion: Enhancing Prompt Understanding of Text-to-Image Diffusion Models with Large Language Models
- [3] VideoDirectorGPT: Consistent Multi-scene Video Generation via LLM-Guided Planning
- [4] GLIGEN: Open-Set Grounded Text-to-Image Generation

---

> ### Author Rebuttal · Authors · 2024-05-31
>
> Thank you for your valuable feedback. Below we address your concerns with further clarifications.
>
> **W1. Paper contributions:**
>
> We would like to point out that the previous works mentioned do not tackle diagram generation, which requires fine-grained/dense layout and relationship control. They also do not use iterative refinement. Below, we summarize our strong contributions:
>
> 1. A new text-to-diagram generation framework that leverages the knowledge of LLMs for planning & refining the open-domain diagrams (Sec. 3), which is more effective than existing T2I methods (Table 1 & 3; Fig. 4 & 5).
> 2. Iterative self-refinement can help create better diagram plans (Table 2; Fig. 13).
> 3. Our diagram plans can be rendered via any vector graphic software (Fig. 6, 15, & 16) and T2I models, allowing for easier uses and human-in-the-loop editing (Fig. 7).
> 4. New dataset (AI2D-Caption; Sec. 4.1) with evaluation suites (VPEval + LLaVA 1.5 finetuned on diagrams; Sec. 4.3) that can help the community benchmark and improve the text-to-diagram methods.
> 5. Improved control of text labels/entity relations (Sec. 3.1).
>
> **W2. Iterative refinement effectiveness:**
> In Table 2, we show that refinement can help improve object relationships (4.56 $\rightarrow$ 4.72). We find the initial diagram plans are mostly correct, and the refinement cleans minor details. Fig. 13 also shows examples of how refinement helps.
>
> **W3. Can LLMs generate diverse diagram plans?:**
> Yes, LLM can generate meaningful variations from the same prompt, as in this **[example link](https://shorturl.at/UpKig)**. We also show that the LLM can generate plans in diverse domains (Fig. 4-7, 10, 12-16). The backbone T2I model is still free to generate the diagrams in any art style, allowing further diversity.
>
> **W4. Visual quality:**
> We agree that the visual appeal of the diagrams can be improved. However, please note that we also proposed diagram rendering via vector graphic software, such as PowerPoint (Sec. 5.4; Fig. 6, 15, & 16). This can create high-quality diagrams that are more visually appealing and easy to adjust (Sec. 5.4; Fig. 7).
>
> In addition, the current backbone is SD v1.4, and we expect that a bigger backbone (e.g., SDXL, SD3) would yield more visually appealing results. However, such scaling is expensive for our academic lab, and we focus on introducing the first LLM-based diagram generation framework. Our methods' flexibility allows adapting to bigger models with sufficient GPU resources.

---

> ### Author Response · Authors · 2024-06-04
> **A Gentle Reminder for Response**
>
> We thank the reviewer for their time and effort in reviewing our paper.
>
> We hope that our response has addressed all the questions and hope that the reviewer can consider revising the score based on our response. We are also happy to discuss any additional questions.
>
> With sincere regards,
> The authors

---

> > ### Comment · Reviewer_Lutp · 2024-06-05
> >
> > Thanks for the detailed response and clarification. I increased my score to 6.

---

### Decision · Program_Chairs · 2024-07-10

**Decision:**

Accept

**Comment:**

The paper proposes DiagrammerGPT, a two-stage framework to generate diagrams from text using LLMs.  The framework first generates a plan (roughly layout of where diagram elements  go) and then generate the diagram from the plan, and adding text labels as a last step.  For plan generation, an iterative refinement strategy using the LLM is proposed.  For diagram generation from the plan, the work uses DiagramGLIGEN - a version of the GLIGEN diffusion model trained on diagrams with captions.  To train DiagramGLGEN, the paper introduces AI2D-Caption, which add captions to an existing dataset of diagrams from AI2 (AI2D) using LLaVA 1.5.  For evaluation and in-context learning, a small subset is further manually annotated.   Experiments against baselines show the proposed framework can generate diagrams that score higher on automated metrics and are also preferred more by humans.

Reviewers are favorable toward the work as the problem it addresses (text-to-diagram) is novel and under-studied.  They also found the paper to be well-written, easy-to-follow, with a well-motivated approach, and convincing experiments and results.  All reviewers are positive, and the AC also agrees the work presents an novel task that would be of interest to the community.  Thus the AC recommends acceptance.

The AC encourages the authors to incorporate feedback from the reviewers and clarifications from the rebuttal for the camera ready, including:
- VPGen analysis (GeX9) and comparison to using GT layout and different LLMs (Bhqq)
- Details for in-context examples, GLIGEN, AI2D-Caption selection, and human evaluation instructions (SF1t)
- Other clarifications and suggestions by reviewers